# Invasions and Local Outbreaks of Four Species of Plague Locusts in South Africa: A Historical Review of Outbreak Dynamics and Patterns

**DOI:** 10.3390/insects14110846

**Published:** 2023-10-31

**Authors:** Roger Edward Price

**Affiliations:** Insect Ecology Division, ARC-Plant Health and Protection, Private Bag X134, Queenswood, Pretoria 0121, South Africa; pricer@arc.agric.za

**Keywords:** South Africa, locust outbreak dynamics, brown locust, red locust, African migratory locust, southern African desert locust

## Abstract

**Simple Summary:**

Four species of plague locust, namely the brown locust, the African migratory locust, the red locust, and the southern African desert locust, have threatened agricultural production in South Africa for centuries. The history of the plague invasions of the red locust and African migratory locust into South Africa are described, as well as the rare local outbreaks of the southern African desert locust in the Kalahari Desert region. The dynamics of the locally produced outbreaks of the African migratory locust in the cereal crop environment on the Highveld are discussed in detail. However, the brown locust is by far the most economically important locust species in South Africa, with a very high frequency of outbreaks occurring in the semi-arid Karoo region. After synthetic insecticides became available in 1945, most of the outbreaks were contained within the Karoo following vigorous control campaigns with only short-lived escapes of swarms into grain production areas outside the Karoo and into surrounding countries. However, the large-scale plague eruptions of the brown locust are almost impossible to stop with the resources currently available to the South African locust control organization, and the threat that the brown locust poses to food security in southern Africa is very real.

**Abstract:**

The current paper provides a detailed review of the historical outbreaks of each of the four plague locust species found in South Africa, namely the brown locust, the African migratory locust, the red locust, and the southern African desert locust. The history and dynamics of the plague infestations and the major local outbreaks are summarized. The typical patterns of the outbreaks of the different species are described, and the threat of these locusts to agriculture in South Africa is defined. The brown locust produces regular outbreaks in the semi-arid Karoo, with large-scale eruptions of plague proportions occurring about once per decade. Patterns of outbreaks often repeat themselves, but the sheer size of the plague outbreaks is almost impossible to stop, and the brown locust has the potential to threaten food security throughout southern Africa. The African migratory locust produces outbreaks in some of the main maize and wheat cropping areas where it is difficult to control. This locust has taken advantage of the man-made crop environment to produce an extra generation per year that was not previously possible in the original grasslands. The coastal area of KwaZulu Natal Province in South Africa was a prime reception and breeding area for plague invasions of the red locust in the past, and the country, therefore, relies on the successful control of outbreaks in east and central Africa to prevent the recurrence of the plague invasions. The southern African desert locust occurs in the Kalahari Desert area, and outbreaks requiring chemical control are rare.

## 1. Introduction

South Africa has four species of recognized plague locusts that have caused economic damage to crops and rangeland grazing, with plagues of some of the species posing a serious threat to agricultural production in South Africa at different times in recorded history over the past 380 years. The plague locusts in order of current economic importance in South Africa are the brown locust, *Locustana pardalina* (Walker), the African migratory locust, *Locusta migratoria migratorioides*, (Reiche and Fairmaire), the red locust, *Nomadacris septemfasciata* (Serville) and the southern African desert locust, *Schistocerca gregaria flaviventris* (Burmeister). All the above species have produced serious outbreaks in the past century that have required intensive chemical control campaigns, while the more historical plague events threatened the food security of the entire country and have formed part of the farming folklore of the country. Various species of grasshoppers are also occasionally recorded as causing economic damage to horticultural and orchard crops in South Africa, especially the elegant grasshopper, *Zonocerus elegans*, and the green bush locust, *Phymateous viridipes*, as well as a number of other species, such as *P. leprosus* and *P. morbillosus, Anacridium moestum* (tree locust), *Cyrtacanthacris* spp., *Oedaleus* spp., and others [1].

Due to the devastation caused to agriculture in South Africa at the turn of the 20th century by incessant plagues of the red locust and brown locust, the South African Government first took over the coordinated responsibility for locust control in 1906 [2]. The first coordinated locust control campaigns were undertaken in 1907 when hand-operated spray pumps and sodium arsenite poison were issued free of charge to the farmers battling the locust infestations. Plague locusts were then declared as national pests in South Africa in 1911 [2], whereby landowners became legally responsible for reporting the presence of locusts on their land and also had to destroy the locust targets using their own farm labor, while the Government supplied the locust poison and spray pumps. However, in various amendments to the Agricultural Pest Act, the Government (Minister of Agriculture) became responsible for the costs of locust control operations, including the labor required for the task, while the farmers still had to report the locusts and assist with the locating of locust targets. This long-standing legal agreement between the farmers and the Government was again ratified under Article 6 of the South African Migratory Pests Act (Act No. 36 of 1983). An updated policy for the management of the locust problem in South Africa was promulgated in 1998 [3].

The issue of sodium arsenite concentrate to the farmers was a logistical problem as it had to be dissolved in warm water and applied as an aqueous spray. The Government’s instructions also advised that sugar should be dissolved into the poison formulation to make the sprayed vegetation more palatable to the locusts. As access to water in the semi-arid Karoo for brown locust control was usually a problem, the farmers were later supplied with sodium arsenite dust formulations that were applied from hand-operated dusting machines, even though the toxicity of arsenic compounds to the spray operators and livestock was already well known in the early 1920s with safe handling and operator protection regulations being issued [4]. The aqueous spray and dust formulations were superseded in 1934 by sodium arsenite mixed with moistened wheat or maize bran as a bait formulation, which was broadcast by hand amongst the roosting hopper bands and adult swarms. The arsenic bait, containing 2–3% sodium arsenite, was more effective and was considered safer for the farmers to apply and was extensively used for the control of brown locust and red locust hopper bands in South Africa during the locust plagues of the 1930s [5].

After the end of World War II in 1945, the organochlorine insecticide, benzene hexachloride (BHC), became available, applied first as a bait ingredient and as an aqueous spray and later as a dust formulation (mainly 6 or 7% gamma isomer in a fine clay carrier, but also as lindane dust with 99% gamma BHC formulation). The BHC wettable powder formulations were applied for decades in South Africa up until the late 1980s from hand-operated or motorized dusting machines at area application rates of 15–20 kg/ha. The organophosphate insecticides, diazinon and fenitrothion, became available in the mid-1970s and were widely used in the field applied as ultra-low-volume (ULV) sprays from a range of motorized mist-blower and stacked-disc sprayers. A standard 400 g/ℓ fenitrothion formulation was applied at a volume rate of 2.5 ℓ/ha, giving an area dose rate of 1 kg a.i./ha [4]. Due to environmental and human health concerns, both the organochlorine and organophosphate products were urgently removed from operational use by the early 1990s and replaced by the synthetic pyrethroid insecticides, deltamethrin UL7 (Decis^®^) and esfenvalerate UL8 (Sumi-Alpha^®^), applied as a ULV spray at a volume rate of 2.5 ℓ/ha from motorized backpacks or vehicle-mounted sprayers), or from aircraft using at 1 ℓ/ha formulation [4]. The vast majority of locust control in South Africa is undertaken using motorized ground equipment [4].

The rationale for undertaking the current review of the historical locust outbreaks is that the patterns of outbreaks tend to repeat themselves, and it is, therefore, vital to understand these historical outbreaks in more detail so that the course of new outbreaks can possibly be anticipated in the future. This is especially important for the brown locust as the regular outbreaks in the Karoo cost many millions of US dollars to contain with chemical control campaigns, and an early warning of how new outbreaks are likely to develop, based on historical experience, could assist with the effective planning and management of future locust control campaigns.

## 2. Materials and Methods

A detailed review of the literature describing invasions and local outbreaks of the four species of plague locusts in South Africa was undertaken. Some of the historical literature dated from the early part of the 20th century and described observations made by visiting mariners to Table Bay of earlier locust invasions, as well as describing the crop losses suffered by the first Dutch settlers in Cape Town in the Cape of Good Hope of South Africa in the 1650s. The history and dynamics of the most serious plague infestations and local outbreaks were summarized from both the historical literature and from more contemporary database records and hand-drawn maps on file at the ARC-Plant Health and Protection (ARC-PHP), as well as from witness accounts, which enabled the typical patterns of the invasions and local outbreaks of the different species to be described. The threat of outbreaks of these four locust species to agriculture in South Africa is defined.

The names of the provinces of South Africa, as well as many of the magisterial districts and town names referred to in the text where outbreaks were historically reported, have since changed under the new political dispensation since 1994. The previous provincial names are recited in the current text to maintain continuity with place names given in the historical literature, but the corresponding modern names of the provinces are given herewith, viz. Cape Colony (the Western Province, Northern Cape Province, and parts of the Eastern Province), Northern Cape (Northern Cape Province), Natal (KwaZulu Natal Province), Orange Free State (Free State Province), western Transvaal (North West Province), northern Transvaal (Limpopo Province), and eastern Transvaal (Mpumalanga Province). The previous names of magisterial districts and towns are mostly retained in the text to aid reference to the literature.

## 3. Results

### 3.1. The Brown Locust, Locustana Pardalina

The brown locust, *Locustana pardalina* (Walker), is the most economically important plague locust species in South Africa. This locust is indigenous to the semi-arid Karoo areas of South Africa and parts of southern Botswana and southern Namibia, with a recognized outbreak area stretching over 250,000 km^2^ of mainly sheep-grazing country across the Nama Karoo biome [6,7]. Outbreaks occur very regularly, with gregarious swarming populations being controlled somewhere or in the Karoo in approximately 90% of the years throughout the entire twentieth century [4]. However, large-scale population upsurges, leading to uncontrollable plague cycles, occur perhaps once per decade, which require a massive chemical control campaign to bring under control [4].

The rainfall gradient across the outbreak area of the brown locust in the semi-arid Karoo ranges from approximately 380–400 mm in the southeast to only 100–130 mm per annum in the arid northwest, with most of the rain falling as thundershowers in late summer and autumn. However, the rainfall patterns are typically erratic and are very patchy in distribution, and extended droughts are common [4]. The relationship between rainfall and the periodicity of brown locust outbreaks has been extensively studied, but no conclusive correlations have been found to date that could aid an early warning system [4]. The brown locust is primarily a grass feeder, and outbreaks pose a direct threat to the sheep grazing rangeland in the Karoo and to cereal crops grown under irrigation within the Karoo, while escaping swarms pose a threat to the main commercial cereal cropping areas in the Free State and North West Provinces [4].

#### 3.1.1. Development of Incipient Outbreaks

The biology and population dynamics of the brown locust are well known [4,6,8,9,10,11,12,13], and the ability of the locust eggs to survive extended dry periods in a state of diapause or quiescence enables egg populations to lie dormant during harsh conditions [14]. Outbreaks often occur in areas where solitary adult populations had been considered as being at a low density for a number of years. Evidently, the egg masses laid by solitary populations can gradually build up in the soil and, under extreme conditions, remain viable over perhaps two to three dry seasons as long as soil moisture and egg turgidity are supplemented by the occasional light rain, to then hatch simultaneously after a wide-spread rainfall event [4,15]. However, there is no evidence that significant egg masses from swarming populations can lie dormant in the field for extended periods of >12 months and have any impact on the outbreak dynamics of the locust [15]. Once the rain falls, the brown locust is capable of rapid population growth under favorable conditions and is multivoltine, producing two to three generations during the warmer months of September to May [10], with four generations recorded under exceptional conditions [4,7]. The behavior of the solitary phase of the brown locust was extensively studied in the field by Smit [10,11], including the life history, population dynamics, the belts of ecotone grass habitats used for concentration by the solitary phase adults for shelter and feeding, the typical mixed grass/bare soil sites used for oviposition and the short-grass carpet habitats favored by the hatching hoppers. Smit [10] recorded a gradual increase in solitary locust numbers over seven consecutive generations, culminating in the formation of localized hopper bands.

Populations of solitary brown locusts in some recession years and during extended droughts sometimes drop to exceeding low densities when it is difficult to find even isolated solitary locusts over vast areas [12,16] (Price, pers. obs.). However, once the rains fall and favorable conditions for breeding return, the populations then rise very rapidly and in synchrony over wide areas. For example, Du Plessis [16] reported the simultaneous population upsurge of solitary locusts across >700 individual farms spread over 13,000 square miles of Karoo and conceded that it was impossible to stop the incipient outbreaks from developing into plagues, with the gregarious swarming populations then persisting for the next 7–11 seasons by exploiting the different summer and winter rainfall areas within the Karoo. Local incipient outbreaks also regularly occur during ongoing plague cycles, being triggered by favorable rainfall conditions, with the new solitary phase populations then gregarizing and forming swarms independent from the main plague-swarming populations present in other parts of the Karoo [16].

Locust transect foot counts undertaken on farms with a known history of outbreaks were used for many years to help predict the local development of outbreaks [13,17]. The adult flush counts on these ‘hot-spot’ farms, which have a history of producing early season outbreaks and which are known locally in Afrikaans as ‘opbouer’ (build-up) farms, gave an accurate indication that outbreaks could be expected and was a valid exercise during the recession periods, but once the widespread incipient outbreaks had started, the counts became redundant. It has also been repeatedly observed that numbers of solitary locusts can still remain high in many areas after a plague cycle of gregarious swarms has collapsed, but these solitarious populations then dwindle for some reason, even if field conditions remain favorable. Lea [12,18] postulated that the genetic ‘quality’ of these solitary locusts, in terms of their density sensitivity or swarming potential, was not the same as the higher ‘potential’ of the solitarious phase locusts at the start of an incipient upsurge across the Karoo. Lea termed the density-sensitive locusts as the “flitters”, which reacted quickly to transform to the gregarious phase over wide areas, while the density-insensitive individuals (the “sitters”) do not readily transform to the gregarious phase and tend to be sedentary and do not disperse, even at relatively high densities. The possible genetic differences or environmental factors influencing the different behavior observed in the solitaria phase locusts is a fascinating theory that still needs to be investigated.

#### 3.1.2. Brown Locust Control Strategy

It is widely accepted that the incipient outbreaks of the solitary phase are impossible to control over a vast area of the Karoo [7,13,16], so it is more practical and economically viable to wait until the locusts aggregate and gregarize and then target gregarious hopper bands and the young adult swarms. The brown locust control strategy that has been in operation since 1907 is upsurge elimination to combat outbreaks at source in the Karoo before migrating swarms can exit the Karoo and threaten the main cereal crop production areas in the Free State and North West Provinces and then further afield. The history of brown locust control in South Africa, along with a detailed review of the strategy and operational tactics employed during control campaigns, was given by Price [4]. As mentioned, landowners and farmers are legally required to report the presence of gregarious locust populations on their land, while the Government is legally responsible for the costs of undertaking the control action. In a large-scale outbreak season, tens of thousands of hopper bands and thousands of adult swarms are individually tracked down and controlled using fast-acting synthetic pyrethroid insecticides applied from a ground-based knapsack and vehicle-mounted ULV sprayers, using an army of temporarily appointed locust officers and labor assistants [4].

#### 3.1.3. Outbreak Frequency

Outbreaks of brown locusts were first reported by European settlers venturing into the Karoo in the early 1700s, with the regular plague cycles reported in the literature from the 1790s, with plagues lasting an average of 13 years and with a quiet recession period of approximately 11 years between the plagues [2,18]. However, Lea [12] observed that since the intervention of chemical control campaigns, which began in 1907, the frequency and duration of plagues with years of great swarming activity and widespread outbreaks, followed by the intervening recession period with only localized outbreaks occurring, were of shorter duration with a periodicity of 7–11 years [12,18]. The rationale for these shorter swarming cycles was that before chemical control intervention, the swarms used to exit the Karoo and swarm and breed throughout southern Africa up to the Zambezi River, leaving the prime breeding areas in the Karoo devoid of locusts for a longer period of time [12,19]. Perhaps the populations of natural enemies also built up to a point where they could exert some natural control on the locust populations [9], while there is current concern that the repeated spraying of locusts in the Karoo has a negative impact on biodiversity and populations of natural enemies [4].

After chemical control was introduced, the gregarious populations were/are mainly controlled within the Karoo, with only relatively minor escapes recorded [4,18]. The strategy of upsurge elimination has certainly prevented any significant damage to crops outside the Karoo but has had the effect of keeping the locust populations contained within the Karoo so that enough swarms survive to lay eggs before winter so that the outbreaks then continue over multiple seasons in different parts of the Karoo. While food security within the southern African region has not been seriously threatened, it is evident that the intense locust campaigns waged in the Karoo on an almost annual basis have failed to prevent the regular cycle of plagues from developing [4,18], which has serious cost and environmental impact considerations.

The historical outbreaks over a 64-year period between 1941 and 2005, based on the number of magisterial districts where chemical control operations against gregarious hopper band or swarm targets was undertaken in the Karoo, showed a variable pattern of outbreak periodicity [4]. If an arbitrary threshold of 10 or more outbreaks, magisterial districts in the Karoo is used as a threshold for the definition of widespread swarming, then high intensity swarming was recorded over an eight-year period between 1946 and 1954, which was followed by a comparative recession period of eight years until 1962. There was a short-lived, intense outbreak between 1962/63 and 1963/64, followed by a recession for a further two seasons until the start of a prolonged period of swarming activity from 1966/67, which lasted 12 years until 1977/78. This was followed by another recession period of about seven years, apart from an upsurge in 1981/82, until the start of the massive cycle of locust swarming that started in 1984/85. The outbreak season of 1985/86 was the biggest recorded since the early 1930s when over 175,000 hopper bands and >40,000 swarm targets were controlled [4]. This period of high locust activity from 1984 continued for 17 years until 2001, although sporadic years within this sequence, such as the 1992/93 and 1998/99 seasons, recorded very low locust activity. It is interesting that these seasons of low locust activity directly followed the intense El Nino drought events of 1991/92 and 1997/98, respectively, over the summer rainfall area of South Africa [20].

Analysis of the outbreak history showed that there was no obvious correlation between locust fluctuations and total annual rainfall or late summer rainfall. However, there was a very clear association that large-scale locust outbreaks occur following widespread drought-breaking rainfall, especially if this rainfall occurred after the comparative failure of the early summer rains [7]. In this regard, the cycles of high locust activity seem to be related in some way to the cycle of El Nino and La Niňa events in South Africa, with La Niňa wet season conditions in the Karoo often associated with the start of intense outbreak cycles, such as the La Niňa event of 1984/85 that drove the massive 1984/85 brown locust upsurge. In contrast, the El Nino events often cause widespread summer drought conditions across the Karoo and a collapse of the more extensive swarming activity. However, the direct correlation between brown locust outbreaks and the pattern of El Nino and La Niňa events, as well as the Southern Ocean Oscillation and even sunspot activity, has yet to be confirmed and still needs further study [4].

The cyclical pattern of brown locust outbreaks continued with a large outbreak recorded in 2006/2007 and another in 2011/2012, which was then followed by a prolonged recession period, possibly due to the influence of intense El Nino events that negatively impacted the summer rainfall region of South Africa from 2013 to the end of the 2016/2017 summer season [20]. However, a wetter rainfall cycle returned from 2018 to date, and between October 2021 and May 2023, South Africa has experienced its worst brown locust outbreak for the past 40 years, with tens of thousands of hopper bands and thousands of adult swarms being controlled. Some migrating swarms even invaded areas close to the south western coast of the Eastern Province and into the Garden Route coastal area of the Western Cape Province for the first time in living memory (Price, pers. obs.). The locust distribution information for this latest outbreak still requires detailed analysis.

#### 3.1.4. Outbreak Patterns

The outbreak area of the brown locust was originally defined by Faure and Marais [21], with the classic high outbreak frequency centers defined by Lea [7]. Maps of the typical outbreak area have changed slightly over the years as rainfall patterns, and perhaps the changing grazing pressure by sheep on the vegetation composition and cover in the Karoo have influenced the suitability of different areas as prime outbreak centers [4]. For example, the most recent map (Figure 1) indicates a shift of the outbreak area further west [22], based on the higher frequency of outbreaks recorded in the northern Bushmanland areas during the good rain cycles from the mid-1980s until the early 2000s, while fewer outbreaks were recorded in the southeastern Karoo during this time. The three areas currently considered by the author as prime outbreak centers include the area between Hopetown-Britstown-De Aar in the eastern/central Karoo, the remote undulating area to the east of Kenhardt towards Marydale and southeast down towards Copperton and Van Wyksvlei in the north-central part of the Karoo, and the desert-grass Bushmanland area southwest of Pofadder in the western Kenhardt District stretching towards Gamoep and Kliprand in the Namaqualand District.

Maps drawn of the seasonal outbreaks of the brown locust, based on the records of where locusts were chemically controlled each week in the Karoo during outbreaks between 1980 and 2005, have provided details of how outbreaks typically develop in different parts of the Karoo (maps on file at the ARC-PHP, Pretoria). History has often repeated itself, and some of the annual outbreak patterns are remarkably similar (Price, pers. obs.).

Based on the prevailing wind conditions at different seasons, it has long been known that locust swarms fledging in the northern western and upper central part of the Karoo in the early summer (November–December) tend to fly in a northerly direction on the prevailing winds [6]. Swarms produced in the southern and eastern Karoo at his time often first fly north and then sharply east when prevailing winds aid their rapid exit from the Karoo into the Free State Province and towards Lesotho. This first generation of young swarms usually fly fast and high over long distances, often at least 100 km per day. As these swarms mature, they slow down and tend to mill around before oviposition takes place. From early February onwards, the second generation of swarms tends to move northeast and east. Later in March and April, the swarms of the late second and the third generation often fly south and southeast to invade the southern parts of the Eastern Cape Province [6]. Typical seasonal outbreak patterns and movements of swarms can be seen from a selection of the locust outbreak maps presented in Figure 2, Figure 3, Figure 4 and Figure 5.

The 1985/86 outbreak, as depicted in Figure 2, started in the central Karoo districts in October 1985, following widespread drought-breaking rains across the Karoo. Uncontrolled swarms escaped and mainly flew north and northwest in December 1985. A widespread second generation developed, and large numbers of swarms migrated north and northwest in February and March 1986 into Botswana and southern Namibia (Figure 2). Later in April and May, swarms flew southeast into the Eastern Cape Province. However, the swarms that had migrated deep into the Kalahari in Botswana in February and March 1986 were able to successfully breed under the warmer early winter conditions found at these latitudes and altitudes, unlike in the Karoo, where winter temperatures are a lot lower and prevent further breeding. The extensive breeding in the Kalahari Desert resulted in numerous very large swarms that migrated back into South Africa between late May and September 1986 on the prevailing winter winds (Figure 3). Many of these flying swarms were controlled by spray aircraft in Botswana, but others migrated into South Africa, where they were controlled in the North West and Free State Provinces. But, some managed to migrate all the way back to the prime breeding areas in the Karoo (Figure 3). This migration pattern possibly gave a glimpse of the historical uncontrolled plague migration patterns of the brown locust within southern Africa that occurred in the days before chemical control intervention at the turn of the twentieth century, with swarms vacating the Karoo and exiting South Africa only for some of their future progeny to reinvade the Karoo in a return migration later during the plague cycle. A total of >175,000 brown locust hopper bands and approximately 40,000 adult swarms were controlled during the 1985/86 campaign [4].

The locust outbreak maps of the 1988/89 and 1989/90 outbreaks show very interesting patterns that highlight the current problem with locust control intervention against the brown locust. A widespread scattered first generation occurred across the Karoo from September to December 1988 and developed into an intense control campaign against the second generation in the Upper Central Karoo and in Bushmanland in the northwestern part of the outbreak region (Figure 4). However, control was not adequate, and hundreds of swarms escaped to produce a third generation that infested a large part of the entire Karoo. From April into May, there was a widespread migration of swarms in a southerly and southeasterly direction, invading the Great Karoo and into the southern and southeastern parts of the Eastern Cape Province (Figure 4). The onset of winter conditions curtailed the outbreak, but not before dozens of swarms had laid egg beds in the southern and eastern parts of the areas invaded. A total of 85,935 gregarious hopper band targets and 12,642 roosting swarms were controlled during the 1988/89 campaign [4].

Although the spring of 1989 was relatively dry, the overwintering nondiapause eggs laid by the swarms had received some winter rain, which ensured widespread hatching on a large scale in the Great Karoo and especially to the south of Aberdeen in the Eastern Province from mid-September until the end of October 1989 (Figure 5). This was a clear example of the challenge facing the locust control organization that despite nearly one hundred thousand gregarious locust hopper band and swarm targets being destroyed during the very intense campaign of 1988/89, enough swarms had survived to lay eggs before winter to continue the swarming cycle into 1989/90. A vigorous control campaign was then undertaken against the hopper bands, but hundreds of young swarms escaped and initially flew strongly in a northerly direction, which was typical for the early summer migrations (Figure 5). However, a weather front in mid-December brought heavy rain and strong westerly winds, and the swarms then flew northeast and then east on the winds (Figure 5). The swarms rapidly emigrated from the Karoo and flew into the Free State Province, with some swarms invading the Kingdom of Lesotho. However, most of the fast-flying young swarms did not have time to mature and lay eggs along their flight path before they exited the Karoo, so there was only a very scattered second generation produced in February and into March 1990 (Figure 5). All the swarms that had exited the Karoo were eventually tracked down and destroyed before any significant crop damage was caused. During the 1989/90 outbreak season, a total of 36,553 hopper bands and 1392 adult swarms were controlled [4].

#### 3.1.5. Threat of the Brown Locust to Agriculture in South Africa

The brown locust produces regular outbreaks in the Karoo, and the large-scale upsurges, if left uncontrolled, certainly pose a threat to agricultural production and the food security of South Africa and the rest of southern Africa. However, upsurges are usually contained within the Karoo with great effort and expense using relatively unsophisticated control tactics and operations. Over the past 80 years, apart from short-term swarm escapes, there have been no significant or wide-scale crop losses due to brown locust infestations outside the Karoo. The ground control tactics utilizing the temporarily employed local farming community whenever there are outbreaks, known as the ‘Commando system’, can be very efficient and cost-effective when dealing with small outbreaks, especially in the eastern Karoo where there is a high population of resident farmers [4]. However, the changing demographics and the depopulation of many farms in the more remote areas of the Karoo, especially in the Upper Central and western Karoo, Great Karoo, and Bushmanland, have negatively impacted the reporting and reaction response to outbreaks in these vast areas so that swarms regularly escape control in these areas [4]. The locust control capacity clearly becomes overwhelmed in these remote areas during the large-scale upsurges by the thousands of locust targets that require control. In addition, the ever-increasing costs of insecticides, spray equipment, labor, and transport cause a budget crisis whenever there are large outbreaks [4]. The long-term sustainability of the current brown locust control system in South Africa is therefore in doubt, and the locust reporting and campaign management system needs to be urgently modernized to utilize available technologies for GIS mapping and for the prediction and real-time monitoring of outbreaks. International support may be necessary in the future to contain the potential menace of the brown locust. Ongoing climate change is also predicted to Increase the frequency of erratic climatic events, possibly leading to more intense drought cycles and floods. The brown locust is known to respond closely to erratic rainfall events in the Karoo, and therefore, the frequency and intensity of outbreaks are likely to be impacted by climate change in the future.

### 3.2. The African Migratory Locust, Locusta Migratoria Migratorioides

The African migratory locust, *Locusta migratoria migratorioides* (Reiche and Fairmaire 1850) is widely distributed throughout the grassland areas of Africa, south of the Sahara. This tropical subspecies of *Locusta* is multivoltine, breeding continuously and without a reproductive or egg diapause [23,24]. The main outbreak area for the African migratory locust (AML) is found in the floodplain grasslands of the Middle Niger River in Mali [1,25], from where swarms developed in 1928 that led to the last great plague cycle that invaded most of Africa’s south of the Sahara between 1928 and 1941 [26]. However, regular swarming on a smaller scale has been reported from at least 12 secondary breeding areas in different areas of Africa [1], but none of these outbreaks has been large enough to initiate a plague cycle. In southern Africa, swarming populations of AML have been regularly reported from the South and Western Provinces of Zambia, especially in the Zambezi Valley and Simalaha plains from 2019 to 2021, the Southeast Lowveld of Zimbabwe (Hippo Valley, Chipinge), the Shire Valley in Malawi, the eastern Caprivi grasslands of Namibia and associated grasslands along the Chobe River in northern Botswana, the Okavango Delta, and around Lake Xai in Botswana and at Simunye in eSwatini (John Kateru, IRLCO-CSA, *pers. com*.). Most of these outbreaks have occurred in natural grasslands and adjacent subsistence and smallholder maize and sorghum farming areas, with serious damage to maize and sorghum crop production regularly reported. Control operations have been undertaken by the International Red Locust Control Service for Central and Southern Africa (IRLCO-CSA) and by local Government migratory pest control officers.

#### 3.2.1. AML Plague Invasion of the 1930s

The last great plague of the AML started in 1928 in the Niger floodplain area [26], from where numerous swarms escaped. Flying swarms first appeared in northern Namibia and Botswana in November 1931 [27], where they laid eggs, and widespread hopper infestations resulted in these areas from December 1931 to February 1932. AML swarms were reported entering South Africa from Namibia in April–May 1932, and fast-flying swarms then flew south-eastwards across the Karoo, covering as much as 240 km per day [27]. Some swarms eventually flew into the sea near East London, while other swarms flew further east into Queenstown and Lesotho. The swarms dispersed during winter, and there was no reported breeding [27]. However, a new influx of swarms invaded South Africa in the summer of 1933–1934, often mixing with invading swarms of the red locust. However, very little subsequent breeding of the AML was reported, with only a few hopper bands produced in the Pietersburg and Soutpansburg Districts in the northern Transvaal (Limpopo Province). Some transient phase AML were also reported with hopper bands of the red locust in the northeastern Transvaal. Some other scattered hopper bands were reported in the Cape Province and Orange Free State, but the swarming populations seem to have died out by the end of 1934 [27]. However, breeding was more successful in countries to the north of South Africa, where the plague ran its course until 1941 [26].

#### 3.2.2. Incipient Local AML Outbreaks

The AML in the solitary phase is found widespread across the subtropical grasslands of South Africa [27] and is sometimes common in irrigated crop areas. After the plague invasions of the early 1930s had died out, there were only rare reports of AML populations causing small-scale damage to wheat. However, from 1960 onwards, the swarming activity was reported on more regular occasions [27]. During the early winter of the 1969–1970 season, a widespread infestation was reported in the Free State (especially the Bothaville, Wesselsbron, Hoopstad, and Kroonstad Districts), with hopper bands congregating along the grassy headlands between the mature maize lands during autumn and into the early winter [27]. Hopper bands and scattered fledgling swarms caused large-scale damage to the winter wheat planted in the vicinity. Numerous dense hopper bands and small but concentrated swarms were produced, which caused alarm amongst the farming community, and the Department of Agriculture had to initiate a vigorous locust control campaign to bring the outbreak under control [28]. Records show that over 4000 hopper bands and nearly 100 adult swarms were controlled using about 50 tons of 7% gamma BHC wettable powder applied from motorized vehicle-mounted sprayers [28].

Then, in 1980, an even larger outbreak of the AML occurred in the same maize-producing areas and spread to a total of 27 magisterial districts, covering a large area across the north-west Free State, North West Province, and into southern Botswana (Figure 6). Thousands of hopper bands and hundreds of small swarms developed in this intense outbreak, which was considered to have almost reached plague proportions [29]. High densities of AML adults congregated on the grassy headlands between the dry maize fields from mid-April onwards and started to fly strongly during the warmer parts of the day in a ‘low streaming flight’, but they did not roost as dense swarms at night. However, in early May, the locusts aggregated into cohesive swarms that roosted at night in dense masses. Numerous swarms of >50 ha in extent were reported that flew strongly during the day [29]. At the peak of the control campaign in May 1980, at least 71 ground-based control teams (applying 7% BHC dusting powder from backpacks and vehicle-mounted machines) were in operation, and the outbreak was only brought under control with difficulty [29]. A spray aircraft and a spray helicopter were also used to control the roosting swarms (mainly applying fenitrothion ULV). Morphometric measurements of the tegmen and femur lengths of locusts collected from a migrating swarm (farm Somerbult, Bothaville District, 31 May 1980) were determined, and the E/F rations calculated as a mean of 1.92 and 1.97 for males and females, respectively (*n* = 30 of each sex), which were deemed typical of the transient phase of the AML, as given by Lean [25].

Further outbreaks occurred at regular intervals for the next decade from 1984 to 1994, with small-scale control operations being mounted almost annually, mainly in 8–13 magisterial districts of the north-west Free State and adjacent districts of the North West Province. Occasional AML outbreaks have also been reported in maize, sorghum, and wheat crops in the ‘Springbok Flats’ and surrounding areas (Potgietersrus, Warmbaths, Waterburg and Thabazimbi districts) of the Limpopo Province (Figure 6), such as the large-scale outbreak in April–May 1993 when >10,000 ha was treated using spray aircraft and vehicle ground teams, mainly using deltamethrin ULV. Outbreaks were also reported at this time in Rustenburg, Marico, and Delareyville districts of the North West Province and in the adjoining Vryburg district in the Northern Cape Province (Department of Agriculture, Directorate: Resource Conservation, Locust situation reports, April and May 1993). The occasional small outbreak has also been recorded in the wheat-growing area of Porterville and Piketberg in the Western Cape Province (Figure 6). However, AML outbreaks in South Africa since 2000 have been less regular, and no significant chemical control has been required for the past decade (John Tladi, *pers. com*.). It is not known why there has only been a low intensity of AML outbreaks over the past 22 years.

#### 3.2.3. Outbreaks of AML in Cereal Crop Environments

Man’s modification of the environment and the seasonal cultivations of crops have enhanced the dry season survival and reproduction of *Locusta* and have caused local outbreaks to occur in a number of new areas around the world [30]. This is especially true in areas of southern Africa where dry season survival is unusually successful, as this allows a greater carry-over of residual populations during the unfavorable dry winter period [30]. Local outbreaks have been recorded in crop irrigation schemes, in areas planted with dryland crops that grow well in winter due to water-retaining soils, or in other favorable habitats that provide green feed and shelter. Likewise, it is evident that the cereal cropland environment in the north-west Free State and adjacent parts of the North West Province on the Highveld of South Africa, where the broad-acre monoculture cultivation of maize in summer is followed by the cultivation of wheat in winter in more limited areas, has evidently favored the local build-up of AML populations [29]. The green feed and shelter provided by winter wheat crops, which are grown adjacent to some of the maize fields, enhance the dry season winter survival and allow the locust to mature and produce an early generation in spring, which was otherwise not possible in the original grassland habitat of these areas [29,31]. The AML was also shown to achieve a higher fecundity in the cereal crops on the Highveld than previously recorded in the main outbreak area of the Middle Niger floodplains. The AML produces two generations per year in the pristine grassland habitats of the Free State Province but at least three complete generations per year in the cereal crop environment. This extra generation and the high fecundity reported in the cereal crop environment enables the AML to produce localized swarming populations by autumn [31].

#### 3.2.4. Population Dynamics of AML in the Cereal Crop Environment on the Highveld of South Africa

The spring wheat generation fledges from mid-November, and following the wheat harvest in early December, this first generation of locusts immediately colonizes the young maize crop that is planted from early summer on a huge monoscale across the Highveld maize belt [29]. The locust densities are very low in the maize at this stage (<5 adults per ha), but numbers build up following successful summer breeding to complete a second generation by February. The hopper populations during summer are of mixed ages as successive batches of egg pods hatch. The locusts then breed again in the green maize, and the third-generation populations can reach peak densities of 3500–5000 per ha in maize fields by April–May (Price, unpublished data). However, during the autumn, the locusts tend to congregate during the day along the unploughed strips of grass headland surrounding each maize field, which is where the aggregation and gregarization of the hoppers often take place and where high adult densities of >10 m^2^ were sometimes recorded. The locusts roost in the maize at night, but after the maize is harvested, the AML populations dramatically crash under the dry and frosty Highveld winter conditions, and very few locusts take up residence in the wheat crop until breeding resumes again in spring. Apart from the 1980 outbreak, there was no evidence of any widespread dispersal of the high-density AML populations at the start of winter, and why higher numbers do not reach the shelter of the adjacent wheat crops is unknown. The peak autumn populations in maize and on the grass headlands, therefore, perish during winter with relatively little carry-over in the wheat until spring, and outbreaks arise de novo each season when conditions are favorable.

The seasonal patterns of the build-up of AML populations in the various habitats in the cereal crop environment on the Highveld were demonstrated by closely monitoring population numbers. Monthly flush counts of adult AML were undertaken in 1984–1986 along 100 m transects walked in maize, grass headlands, and wheat fields at a series of farms with a history of previous AML outbreaks in the Free State and North West Provinces. The number of adult locusts flushed up immediately in front of the walker over a 1 m width were counted using hand-held tally counters along the 100 m transects walked at random in the seasonal maize and wheat crops. The counts along the narrow grass headlands were undertaken using the same path each time. Counts were only undertaken when it was warm enough for the locusts to fly up from the ground. A total of 10 transects were undertaken in each habitat at each visit, and the mean flush counts were recorded (Price, unpublished data). An example of a data set of flush counts for seasonal maize and wheat crops and along the permanent grass headlands is shown for the farm Kromvlei (27°27′ S, 26°45′ E) in the Bothaville district of the Free State Province between February 1984 and December 1986 (Figure 7, Figure 8 and Figure 9). The mean flush counts of AML adults are shown on a logarithmic scale as counts along the grass headlands reached >500 per 100 m transect at peak outbreak season. During the peak season months of March–April, additional visits were sometimes made to the farm to undertake an additional AML count in maize and grass headland habitats.

AML population numbers in maize fields on the farm Kromvlei rose rapidly in late summer and autumn, reaching a peak in April of each year (Figure 7). Maximum densities in maize were 0.3–0.5 AML per m^2^. Locusts flew spontaneously during the day, but there was little evidence of any dispersal at night, with locusts found static in the maize fields even on relatively warm nights, and only the very occasional AML was found at street lights in local towns during peak summer. The maize crop was usually harvested in June, and population numbers then dropped dramatically. Populations along the grass headlands peaked between March and June as locusts made a daily move from their roosts in the drying maize crop to congregate during the day along the open grass habitat to bask and feed (Figure 8). Peak densities of 5–10 locusts/m^2^ were recorded on some occasions. The grass headlands were also where hoppers tended to aggregate and where phase transformation took place. Although scattered hopper bands were common, the E/F ratios of adult locusts collected during peak populations in 1985 measured 1.87 and 1.84 for males and females, respectively (*n* = 30 of each sex), which were considered typical for the solitarious phase, as given by Lean [25]. The wheat habitat was seasonal, available from May to December each year, and AML adults first colonized the young wheat crop in June–July. Populations bred in the wheat between August and September, but adult numbers were always relatively low (Figure 9). Hopper populations were recorded between September and November, but fledglings rapidly left the crop when it was harvested in December.

#### 3.2.5. Threat of the African Migratory Locust to Agriculture in South Africa

By the time the high-density AML populations are achieved in autumn (March–May), the maize crop is mostly ripe and drying out for harvest, and the AML does not usually pose a threat to most of the maize crop. However, some small-scale economic damage has been observed in late-planted maize where the developing cobs have been damaged, and the AML populations at high densities can also strip the maize leaves that would have been utilized by the farmers as winter fodder for stock animals [32]. The main economic damage is caused by newly planted wheat that is grown adjacent to standing maize fields. The high AML populations roosting in the dry maize crop in June and July make daily feeding forays into the adjacent newly emerging wheat crop boundary areas where widespread damage and crop loss have been reported on occasion [29]. In outbreak years, there are reports of farmers having to replant their wheat crops. The AML is difficult to control in the cereal crop environment, and outbreaks can arise relatively unnoticed, so the AML will continue to provide a low-level threat to crop production under current agricultural practices, but with only the occasional outbreak causing economic concern. Changes to agricultural practices to deny the AML an overwintering or dry season habitat have been advocated as a possible management tool for tropical *Locusta* [30].

### 3.3. The Red Locust, Nomadacris Septemfasciata

The recognized outbreak areas of the red locust, *Nomadacris septemfasciata* (Serville), occur in eight relatively small, seasonally flooded, and remote grassland areas in central and east Africa [33,34,35]. A few other small-scale subsidiary outbreak areas are also recognized in eastern and southern Africa, where high-density populations are known to occur [35]. Likewise, outbreaks that produced hopper bands and swarms are also known to occur in Madagascar and in the Lake Chad and River Niger areas [1], but these areas are not considered as plague source areas. The biology and population dynamics of the red locust and the ecology of one of the main outbreak areas in the Rukwa Valley in Tanzania have been extensively studied [1,36,37,38,39,40]. Three major red locust plagues are recorded in the literature [1], with the last great plague lasting 18 years from 1927 to 1944 and invading most of Africa, south of the equator, except for the more arid southwestern areas [33,41]. Following this devastating plague cycle that inflicted serious damage to crops and pasture and threatened the food security of entire countries, it was clearly evident that individual countries could not tackle the threat of the red locust alone, which eventually led to the establishment of the International Red locust Control Service (IRLCS) in 1949. The IRLCS convention was signed by nearly every country that had suffered infestations by the red locust and had the primary objective of controlling locusts in their outbreak areas to prevent the recurrence of plagues. The history of the IRLCS and its subsequent development into the IRLC0-CSA in 1970 has been described in detail by Byaruhanga [34].

As the red locust is univoltine species and only breeds consistently and successfully in its seasonally flooded grassland outbreak areas, which occupy a relatively small area of less than 1500th the size of the invasion areas [41], the red locust is therefore vulnerable to targeted chemical control operations in the outbreak areas. The last major plague ended when the first organochlorine insecticides became available in the 1940s [33]. Since then, the numerous insecticide interventions undertaken against hopper bands and against swarms escaping from outbreak areas at various times have contained outbreaks before any new plague cycles could get going [33,35]. However, population upsurges are often difficult to detect in the remote and often inaccessible outbreak areas, and once swarms have escaped from these source areas, they become highly mobile and are difficult to track down and control in the vast invasion area. The red locust occurs in the solitary phase in the subtropical grassland areas of South Africa in northern Zululand and in the Kruger National Park, with isolated specimens also recorded from cultivated areas in other parts of the country [42]. For example, the red locust is occasionally observed in maize and wheat crops on the Highveld, with isolated egg pods being found between the maize rows (Price, pers. obs.). However, no incipient swarming populations have been produced in South Africa outside of the recognized plague cycles.

Lounsbury [2] and Lea [18] described reports of red locust infestations in South Africa from 1840 to 1853 and then again from about 1888 to 1907, while the third plague invasion between 1933 and 1944 is described in great detail [18,43,44]. Apart from these three recognized plague invasions, Lounsbury [2] also reports earlier accounts of locusts ravaging the gardens of the Dutch settlers in the Cape Town colony in 1653, in 1687, and again in December 1746. It is probable that these invasions were also of the red locust as records state that the locusts ate the leaves of the trees, which is not typical feeding behavior of the indigenous brown locust (Price, pers. obs.). In addition, Lounsbury [2] states that even in brown locust plague years, the brown locust swarms did not reach the Atlantic coast of the Union of South Africa.

#### 3.3.1. The First Recorded Plague Invasion

Lounsbury [2] reviewed local historical literature that recorded the red locust as a serious problem in the Cape Colony in the 1840s, where the local farmers knew it as a different locust species compared to the more familiar outbreaks of the brown locust in the Karoo. Lounsbury [2] records that Cape Town was invaded in February 1843, and crops, vineyards, and pastures were greatly damaged while enormous numbers of locusts flew out to sea and later washed up on beaches around Cape Town. From 1847 to 1853, the red locust infestations also became a serious pest problem in Zululand in the Natal Colony, where oviposition was regularly reported from October to December during the years of infestation. However, in the Cape Province, under cooler climatic conditions, the swarms only lay eggs in February and March [2]. Hoppers were, therefore, regularly reported from October to June all along the coast from Zululand to the southern Cape. Even in those early days, it was reported that the red locust only bred once per year, in summer at the start of the rain season and that no second generation of hoppers was produced.

#### 3.3.2. The Second Plague Invasion

During the second plague cycle from 1888 to 1907, Lounsbury [2] recorded that “great swarms flew out of the Kalahari” and that in 1893, red locust swarms arrived in the Natal Colony. In early 1895, swarms were reported in Griqualand West and along the border of the Northern Cape, as well as along the Transkei coast. In late 1895, large numbers of swarms invaded the east of South Africa (presumably from Mozambique) and migrated along the Natal coast, while swarms invading from the north from Botswana crossed the Karoo on a southeasterly flight path, passing to the east of Carnarvon and Victoria West before they all tended to converge on the coast near Port Elizabeth (Gqeberha) [2]. The converging swarms then continued westwards in a broad belt along the coast before diminishing along the southern Cape coast near Robertson and Swellendam by March 1896. Swarms did not persist in the southern Cape and died out. However, other swarms remained along the coast eastwards from Port Elizabeth for a number of years thereafter, becoming particularly prevalent in northern Natal. Populations gradually retreated to northern Natal and to parts of the Lowveld area of the eastern Transvaal (Mpumalanga Province), as well as in Swaziland (eSwatini) and in southern Mozambique. Lea [18] records the plague cycle as ending by the end of 1907.

#### 3.3.3. The Third Plague Invasion

The migration patterns and breeding records of the red locust during the last plague cycle in Africa between 1927 and 1945 have been systematically described by Morant [41] when migrating swarms rampaged across almost all of central and southern Africa. Red locusts were first reported in South Africa in 1933, although swarms were reported in both Botswana and Namibia from 1930 to 1931 [43]. Fascinating details of the progression of the red locust invasions and subsequent breeding in South Africa are given by Du Plessis [43,44]. Invasions of the red locust into South Africa began during the 1933–1934 summer season, and by the winter months of 1934, numerous swarms were reported over almost the entire eastern parts of the country, as well as into northern Natal. According to Morant [41], this invasion was by swarms of the seventh generation of the plague cycle that congregated in the northern Natal coastlands, which then became the second most important center for successful breeding and for the production of new swarms for the remainder of the plague cycle after the prime Zambesi focus area [41]. From November 1934, a large new wave of swarms of the eighth generation of the plague cycle invaded the Transvaal from the north of the Bechuanaland Protectorate (Botswana) and also from Mozambique into Natal [41]. By January 1935, the swarms had reached the southeastern Cape coast at Humansdorp. Other swarms entered the southern Orange Free State and flew into Basutoland (Lesotho). The main swarm incursion was recorded in November and December 1934 when there was great national concern that agricultural production would be threatened [43]. Oviposition started in November 1934 in the eastern Transvaal, but then wide-spread oviposition was reported all along the eastern coast from December 1934 into January 1935. Oviposition in Port Elizabeth and Humansdorp only started in January 1935. Hatching started from late November into December 1934, but severe late summer drought conditions in South Africa from the end of January until March 1935 must have desiccated the egg deposits, and only minor hopper outbreaks were reported [43]. By April 1935, only a few hopper bands and small swarms were still being reported in the Orange Free State and in the Eastern Cape Province. It was evident that the drought had broken the plague cycle in South Africa. However, some other swarms entered Natal from Mozambique in March 1935, but little damage was caused [43]. These swarms were controlled by poison dusting from aircraft in northern Natal, whereby fine sodium arsenite dust was simply released into the slipstream of a modified aircraft (the three-engine Hercules) over the roosting swarm targets [28].

In September 1935, a few swarms were again reported in northern Natal, with one or two swarms penetrating as far south as Durban. A few swarms were also reported in the Soutpansburg District in the Transvaal. In October 1935, swarms were again reported in northern Natal, with some penetrating further south along the Natal coast to the Port Shepstone area. From October to November 1935, swarms from the Transvaal flew southeast and entered Natal, with massive oviposition occurring along the coastal areas of Zululand from mid-November and into January 1936 (ninth generation of plague cycle [41]). However, drought conditions caused this generation of hoppers to fail over most of the infested area with few swarms produced, but breeding was more successful in the northern parts of Natal, and a new generation of swarms resulted from April 1936 onwards [43]. During the 1936–1937 season, the hopper outbreaks were on a more extensive scale than occurred during the previous 1935–1936 season, and hatching continued over an extended period, especially in the Natal foci area. However, heat and drought reportedly destroyed egg deposits in the western areas of the Cape, as well as in Botswana [44]. Poison bait was now being used on an extensive scale with good results against hopper bands. Numerous flying swarms were reported in September 1936, flying south and southeast, caused by a large southerly prebreeding migration of swarms into Rhodesia and the eastern parts of South Africa and into Natal [41]. The migrations occurred until December when the swarms became stationary while they oviposited over a vast area. Breeding was successful, and the resultant swarms tended to congregate along the Natal coast, where they then drifted up and down parallel to the coast as had occurred previously [41]. After September 1937, there was yet another invasion of South Africa, but there were only small outbreaks reported into 1938–1939, and for the first time since 1933, no swarms were produced in the Natal coastlands [41]. In September 1939, there was again an influx of swarms into the Natal coastlands, and breeding occurred on a fairly large scale. The South African locust control authorities conducted intensive control operations with sodium arsenite bait against the hopper bands and with the aerial application of sodium arsenite dust against the roosting swarm targets in the trees. Series of specimens collected from the migrating swarms of red locusts in Pretoria and at locations in the northern and eastern Transvaal and northern Natal from 1936 to 1940, which are housed at the National Collection of Insects at the ARC-PHP, exhibit the extreme gregaria phenotype with a constricted pronotum, smaller body size compared to the solitarious phase, and being a deep red/chocolate brown color (Price, pers. obs.).

The hopper generation in 1940 was very limited and scattered. During the winter of 1941, a few swarms of local origin were again reported along the Natal coastlands, followed by another influx of swarms in October and November that migrated southeast across Botswana into northeastern parts of South Africa, but only localized breeding from these swarms was reported in Natal into 1942. Large migrations of swarms were again reported after September 1942, flying across Botswana and the northern Transvaal to reach the Natal coastlands. Hopper infestations in 1943 were confined to the Natal coastal foci area, but no swarms were produced following intensive control operations. Small-scale infestations were evident in early 1944 in Natal, but all hopper bands were destroyed, and the plague cycle in South Africa had ended. The red locust plague across Africa had finally died out by early 1945 [41].

#### 3.3.4. Small-Scale Red Locust Outbreaks in Southern Africa

After the last plague subsided, the red locust activity in southern Africa remained very calm, with only infrequent population upsurges and slight damage to crops reported [1,45]. However, red locust swarms invaded Zimbabwe in 1972 from the Buzi-Gorongosa floodplains in central Mozambique, and some 16 swarms were controlled, leading to the recognition of the Buzi-Gorongosa as a main outbreak area. Zimbabwe was again invaded in 1974, and a red locust swarm was reported in Malawi in 1978. A high-density adult swarming population was reported and controlled in sugarcane at the Simunye Sugar Estate in northeastern Swaziland (eSwatini) in August 1982, where approximately 100 ha of infestations were controlled by a spray helicopter sent by the South African Department of Agriculture upon request from the Swaziland authorities [45]. This was the first reported swarm concentration reported south of the Limpopo River since 1944 [45]. High-density populations were again reported from the same Simunye Sugar Estate in May 1984, which were controlled over an area of approximately 60 ha using a spray helicopter from South Africa, using an aerial application of 40% fenitrothion at an area application rate of 2 ℓ/ha.

However, in 1996, a widespread hopper outbreak was reported in the Buzi-Gorongosa outbreak area in Mozambique. Hopper bands were first reported in December 1995 at the Mafambisse sugar estate on the northern bank of the Pungue River, but widespread hopper bands were reported in the natural floodplain grasslands from January to March 1996 [46]. Fledglings were first reported in February 1996, and dense swarms formed from March to April. Despite ongoing control operations, numerous fast-flying swarms then escaped the outbreak area, with some swarms migrating large distances to cross international boundaries to invade Zimbabwe, Zambia, Botswana, Malawi, and South Africa [46]. The swarm outbreak was on a scale not seen in southern Africa since the great plague of 1929–1944, with some of the largest swarms ever seen by staff of the IRLCO-CSA [46]. However, the only successful breeding achieved by these swarms was reported in Malawi [46].

The migration routes of the swarms escaping from the Buzi-Gorongosa outbreak area in 1996 are given by Bahana and Ngazero [46]. Most of the escaping swarms flew north into the Tete, Manica, and Zambezia Provinces before invading Zimbabwe to the west and Malawi to the north. From Zimbabwe, the swarms rapidly flew across the country and entered Botswana and Zambia. At least two swarms then flew south and entered South Africa from Botswana, with one swarm being controlled by spray aircraft to the north of Pretoria in October 1996. There was also a report of another swarm sighting, but this swarm later dispersed somewhere near Pretoria. This was the first invasion by gregarious swarms of the red locust into South Africa since the last great plague invasion of the 1930s.

Extensive swarm control operations continued in the Buzi-Gorongosa outbreak area against the coalescing and milling swarms until late November 1996 using spray aircraft. Only small-scale hopper bands and fledgling swarms then resulted in the grassland plains in early 1997, along with small isolated outbreaks occurring along the previous flight path of the swarms into central Mozambique [46]. The conclusion was that the Buzi-Gorongosa outbreak area has the high potential to produce a substantial number of swarms that can easily escape control and initiate a new plague [46].

#### 3.3.5. Threat of the Red Locust to Agriculture in South Africa

The northern Natal coastal area of South Africa was a reception area for migrating swarms of the red locust during the last plague cycle and became a major center for successful breeding and for the production of numerous swarms that drove the plague cycle in southern Africa over an extended period [41]. Although local incipient populations of the red locust in South Africa seem incapable of building up to swarming populations by themselves, the Natal coastlands are a prime breeding area for invading swarms, and the potential threat of the red locust was clearly seen during the brief swarm invasion in October 1996 from the Buzi-Gorongoza outbreak. South Africa, therefore, relies on the effective control of upsurges in the main outbreak areas in central and eastern Africa by the IRLCO-CSA and associated Government control campaigns in the outbreak countries in order to prevent new plagues from developing in southern Africa and the resultant invasions into South Africa.

### 3.4. The Southern African Desert Locust

The southern African desert locust, *Schistocerca gregaria flaviventris* (Burmeister, 1838), is recognized as a subspecies of the well-known plague desert locust, *Schistocerca gregaria* (Forsk, 1775) that ranges over a huge area of North Africa, the Arabian Peninsula, and into Southwest Asia [47]. The southern form of *Schistocerca* is geographically isolated in southern Africa by the tropics and a wide belt of bush or woodland [48], and although it can interbreed with the northern desert locust in cages, the offspring are infertile [48,49]. In addition, the behavior and morphometric measurements of the two subspecies are also very different [48]. Recent studies have indicated that the level of genetic diversity of *S. g. flaviventris* was moderately lower than in the northern subspecies, although the separate populations were genetically homogeneous, such as observed in the northern subspecies [50]. In addition, univariate and multivariate morphometric analysis of the northern and southern populations of the desert locust indicated morphological differences that suggest that the southern population may be categorized as an evolutionary dichotomy and may separate the two locust populations into two distinct species [51].

#### 3.4.1. Distribution and Population Dynamics

Outbreaks of *S. g. flaviventris* occur mainly in the Kalahari sand-dune region of southeastern Namibia, southern Botswana, and in the Gordonia district of the Northern Cape Province in South Africa, where outbreaks sometimes coincide with that of the brown locust, causing confusion with identification [52,53]. The solitary phase of *S. g. flaviventris* has a very wide distribution over the semi-arid and arid areas of South Africa, Botswana, and Namibia [48,49], with some individuals recorded along the coastal belt in southwestern Angola and even on the Ascension Islands [47]. In South Africa, solitary phase adults can be regularly found in the western coastal districts and also in the arid areas of southern and southwestern Namibia [52]. The locust is less frequently found across a wide area of the Bushmanland and Nama-Karoo areas of the Northern Cape Province, although it is sometimes common in cultivated lands in the Calvinia, Kenhardt, and Hopetown districts [52,53]. Hoppers are rarely found outside of the Kalahari sand-dune region on the southeast border area between Namibia and South Africa (Gordonia district border).

The biogeography and biology of *S. g. flaviventris* has been reviewed by Walloff and Pedgley [49]. The locust thrives in years of good desert rainfall but can survive in extremely dry areas (<100 mm annual rainfall). Female *S. g. flaviventris* mature in spring and only oviposit in moist sandy soil after good rains have fallen [19]. Under favorable rainfall conditions, two generations can be produced per year. The female lays 2–3 egg pods in her lifetime, with solitary females laying 95–158 eggs per pod, while more transient/gregarious females lay smaller clutch sizes with <80 eggs per pod [19]. The average time taken for the eggs to hatch is 14–27 days, depending upon temperature and soil moisture conditions [19]. Two generations are possible per summer rain season in the field. Adults overwinter in an immature state and disperse widely. There are five hopper instars, with an average total hopper life of about 60–129 days in summer and autumn conditions, respectively, in outdoor cages in Pretoria [19]. Adults overwinter in reproductive diapause. Solitaria phase adults are very strong fliers and migrate at night over long distances during summer [53] and can be occasionally found attracted to street lights in towns throughout the northwestern and central areas of the Northern Cape Province and in the Namaqualand region of the Western Province (Price, pers. obs.).

#### 3.4.2. Outbreaks of the Southern African Desert Locust

The southern African desert locust rarely reaches swarm status, with only two swarming outbreaks being recorded in the literature (May 1934 and March–April 1948) in South Africa during seasons of exceptionally good rains in the desert areas [48]. In 1934, approximately 100 concentrations of fledgling adults were controlled in southern Namibia, and some invaded South Africa in the winter of 1934 [53], where they caused damage to citrus trees and other crops. A small number of swarms were later reported at Calvinia and even at Worcester in late 1934, while some hopper concentrations were later controlled in the Calvinia, Kenhardt, and Prieska areas in 1935 [53]. In 1948, a total of 469 incipient hopper bands were controlled from March to April, along with 4 swarm concentrations [28,53]. Hopper bands were controlled in 1951 and again in 1964 [53], while hopper bands and at least 25 fledgling concentrations were controlled from February to April 1994 (Department of Agriculture: Directorate Resource Conservation locust control records, February to April 1994). In 2011, reports of aggregations of solitary adults were received from a farm in the Calvinia district, which is well outside the recognized prime breeding area, but no control was undertaken. The current National Department of Agriculture (Directorate: Climate Change and Disaster Management), which is responsible for locust control operations, has no records of any desert locust control operations for at least the past 10–15 years (John Tladi, Directorate: Climate Change and Disaster Management, *pers. com*., June 2023). Chemical control of hopper bands and small swarms has also been undertaken during outbreak years in the Aroab, Karasburg, and Warmbath areas of the eastern Karas administrative region of southern Namibia. Interestingly, outbreaks have also been reported on a number of occasions further west at Aus (Luderitz District) and in the western Karasburg District on the edge of the Namib Desert.

Solitary phase hoppers are usually a uniform light green color, while transient phase hoppers become more yellow and orange in color, with some having black/brown stripes. A more gregarious phase has been reared under crowded conditions in cages where they become more striped black in appearance with yellow/orange spots and lines, but it is unlikely that this form has been seen in the field [52]. Localized concentrations of transient phase hoppers occur perhaps once or twice per decade after good rains in the Kalahari sand-dune region when scattered bands can be observed marching along the dune ‘streets’ between the dunes, but these localized outbreaks are considered as being of novelty interest only as the local farmers are aware that they are unlikely to cause any damage. On occasions when adult swarms have formed, they reportedly kept close to the ground in a loosely aggregated ’butterfly’ formation and did not fly long distances [53].

#### 3.4.3. Threat of the Southern African Desert Locust to Agriculture in South Africa

Migrating adult swarms have occasionally been recorded in the past, invading irrigated farming areas along the Orange River and damaging crops such as citrus, cotton, tobacco, vines, and vegetables [53]. However, outbreaks that are big enough to pose an economic threat are very rare, and the threat of this locust to agriculture is considered low.

## 4. Discussion

Outbreaks of plague locusts in South Africa have been recorded for centuries and are a common sight amongst the farming communities, especially in the semi-arid Karoo. The brown locusts produce regular outbreaks in the Karoo, but the location and intensity of the outbreaks vary greatly between years. Most of the outbreaks over the past 70–80 years have been adequately contained within the Karoo by the South African locust control organization, and swarm escapes into cereal cropping areas outside the Karoo and into surrounding countries are rare. Of concern, however, is that the traditional community-based ‘Commando system’ of tracking down and spraying individual locust targets using vehicle-based ground control teams has become ineffective in the more remote areas of the Karoo, especially during plague outbreaks when the sheer scale of the task of having to control tens of thousands of hopper band and swarm targets sometimes overwhelms the locust control resources. In the opinion of the author, the threat posed by the brown locust to agriculture and food security in southern Africa will only increase unless the South African locust control organization is able to modernize and increase its ability to tackle the plague cycles. As previously discussed by Price [4], the locust control organization needs to urgently introduce electronic reporting and mapping technologies in order to support effective campaign management, as well as the judicious use of spray aircraft in the more remote areas of the Karoo to target the young swarms as they coalesce into large swarm targets. More research is also required to understand the outbreak process in relation to climatic conditions in order to provide a better early warning system. The frequency and intensity of outbreaks are also likely to change with climate warming as the brown locust outbreak dynamics closely respond to periodic droughts and wet cycles, which may become more erratic in the future.

The African migratory locust poses a threat as incipient outbreaks occur directly within the vulnerable cereal crop environment. Outbreaks can also be difficult to control in the cropland habitat, but fortunately, the harsh winter conditions on the Highveld cause outbreaks to rapidly collapse. However, the growing of cereal crops in more temperate areas where overwintering survival may be enhanced could cause future problems with this locust. The red locust is difficult to control in its remote and seasonally flooded grassland outbreak areas. Fast-flying swarms can then easily escape and invade a vast area of southern and central Africa. South Africa relies on the efficient control function provided by the IRLCO-CSA, and the threat of swarm invasions into South Africa over the past 75 years has been rare. South Africa is not currently a member of the IRLCO-CSA, with the proviso being that the red locust has not provided a notifiable threat, and South Africa is already controlling the outbreaks of the brown locust to protect the rest of southern Africa. However, the red locust still has the potential to swarm on a vast scale if the monitoring and control guard is dropped. The southern African desert locust does not readily swarm and has only produced small outbreaks of novelty value. However, the desert locust responds quickly to exceptional rainfall seasons in the southern Kalahari, and the frequency of such events may increase with climate change in the future.

## Figures and Tables

**Figure 1 insects-14-00846-f001:**
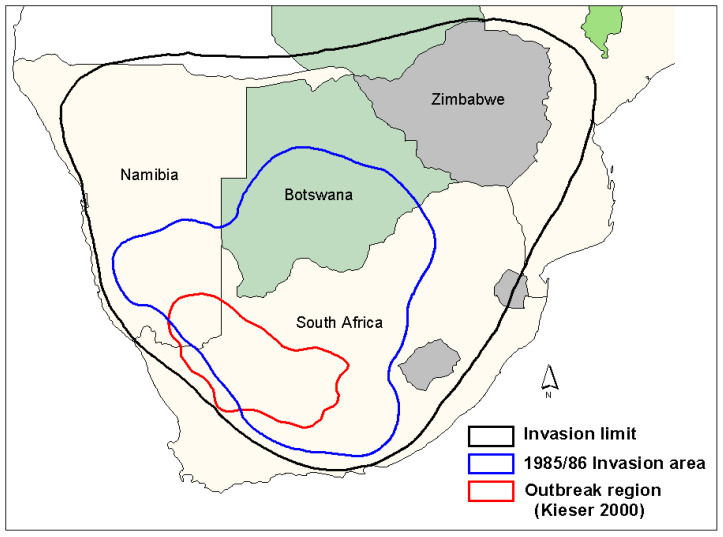
Revised map of the outbreak region of the brown locust, along with the invasion area of the 1985/86 upsurge and the invasion limit as defined by Lea [7].

**Figure 2 insects-14-00846-f002:**
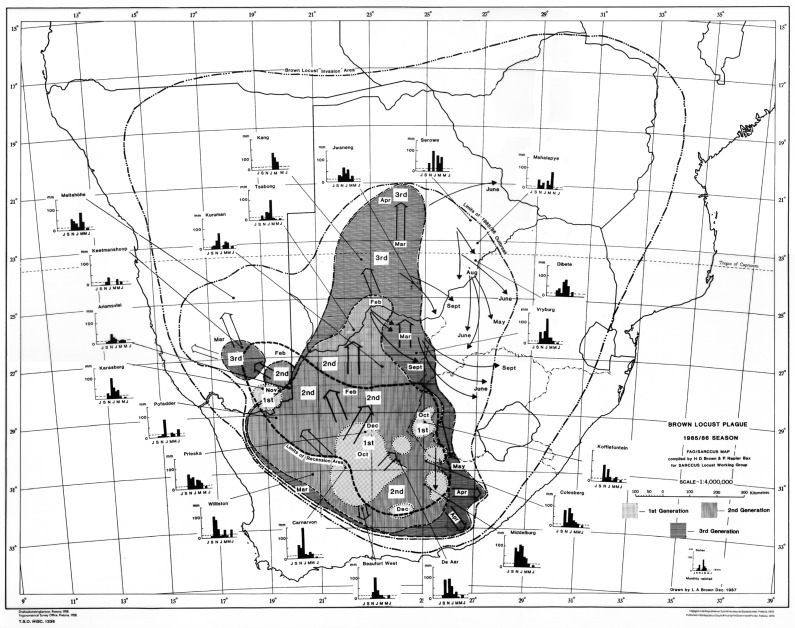
The development and extent of the unprecedented outbreak of 1985/86 show the generalized direction of swarm movements at different times of year. The boundary limit of the 1985/86 outbreak shows the limit of the recorded migrating swarms. Rainfall records for various towns across the outbreak region are included.

**Figure 3 insects-14-00846-f003:**
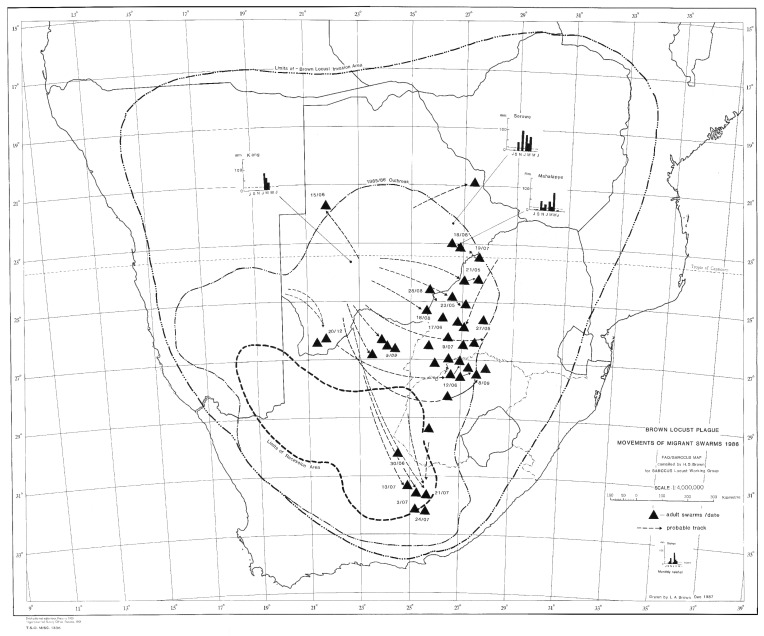
Brown locust swarm migrations from Botswana into South Africa during winter of 1986.

**Figure 4 insects-14-00846-f004:**
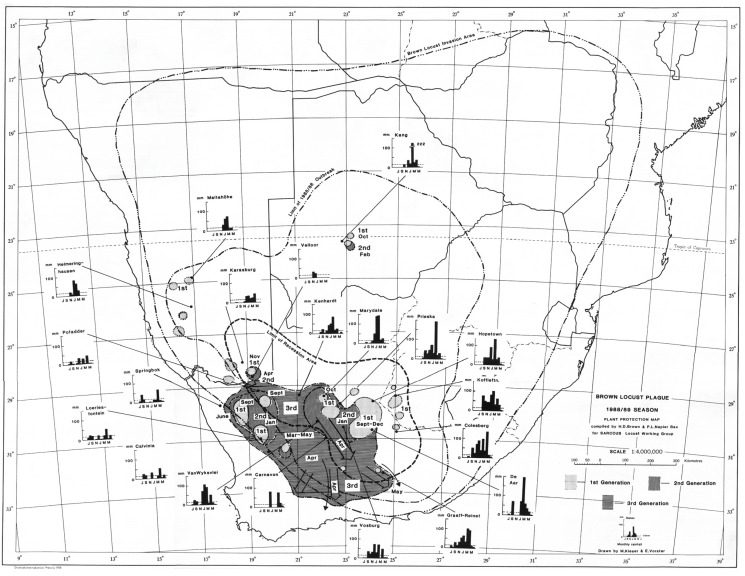
Map of the brown locust outbreak for the 1988/1989 locust season, showing main movement of swarms in autumn. Rainfall records for various towns across the outbreak region are given.

**Figure 5 insects-14-00846-f005:**
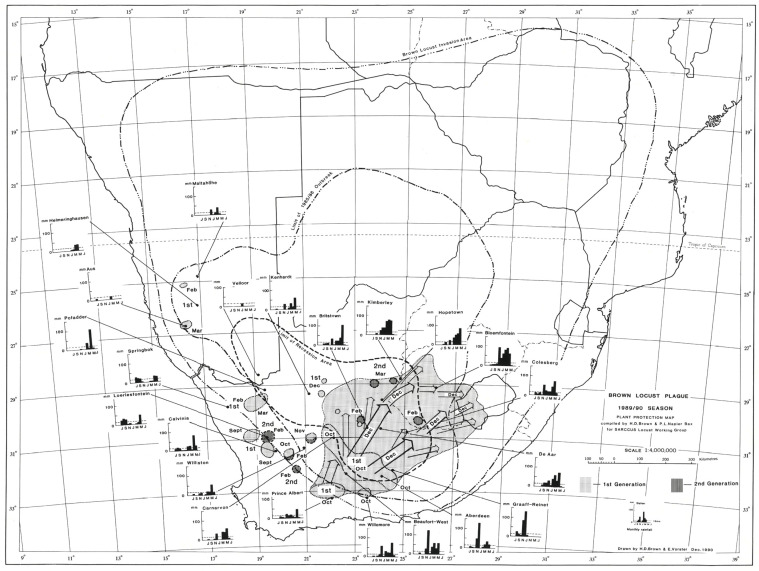
Map of the brown locust outbreak for the 1989/1990 locust season, showing main movement of the swarms in summer. Rainfall records for various towns across the outbreak region are given.

**Figure 6 insects-14-00846-f006:**
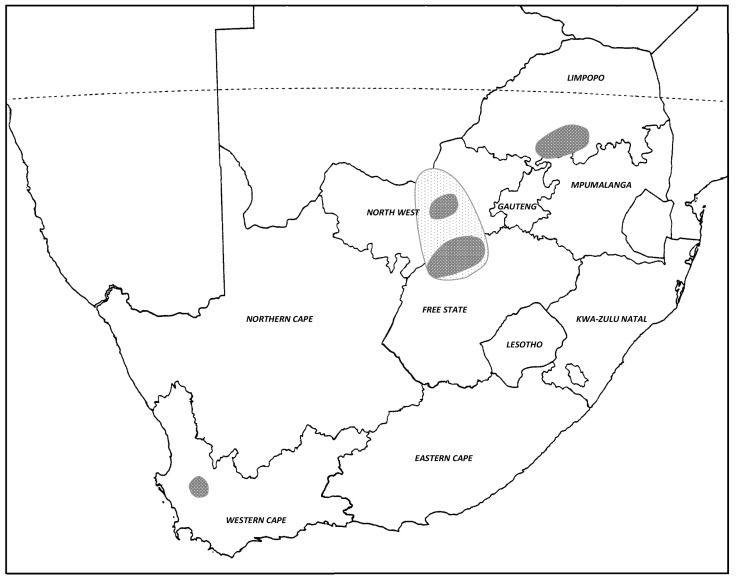
Areas of South Africa where outbreaks of the African migratory locust have required chemical control intervention. The limits of the 1980 outbreak in the Free State and North West Provinces are depicted.

**Figure 7 insects-14-00846-f007:**
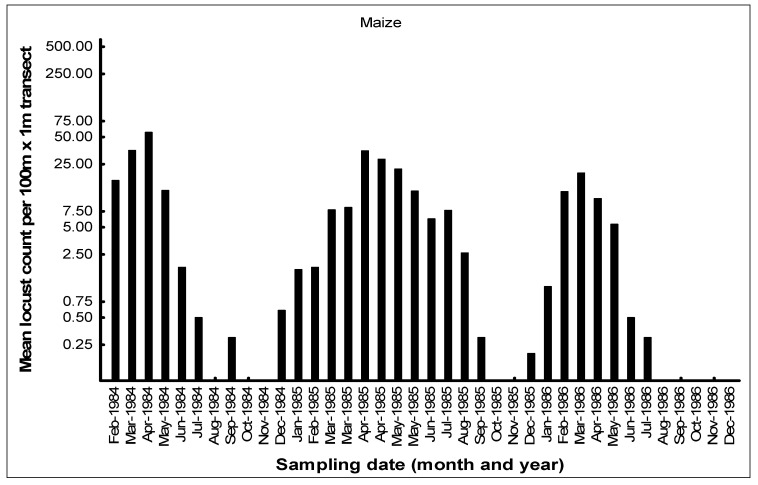
Mean flush counts of AML adults along 100 m × 1 m transects (*n* = 10) in maize fields on the farm Kromvlei, Bothaville district, between February 1984 and November 1986. Months with zero locust counts are shown as dots. Months when no counts were made have no data.

**Figure 8 insects-14-00846-f008:**
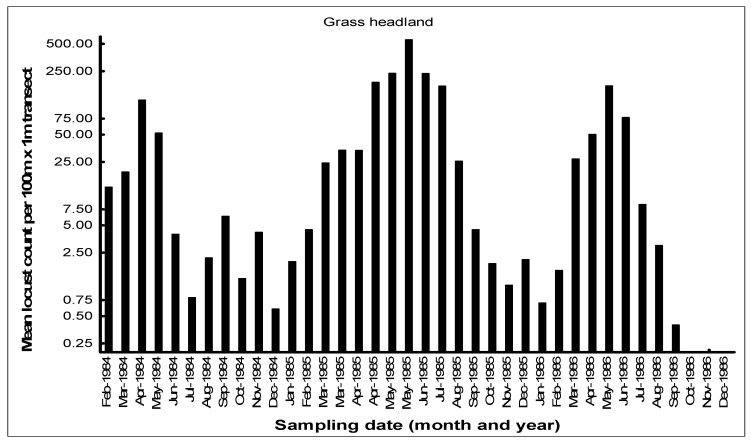
Mean flush counts of AML adults along 100 m × 1 m transects (*n* = 10) walked along grass headlands between maize fields on the farm Kromvlei, Bothaville district, between February 1984 and November 1986.

**Figure 9 insects-14-00846-f009:**
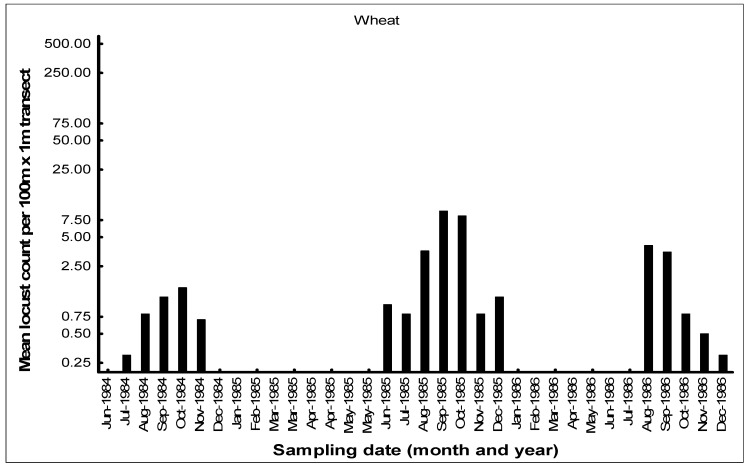
Mean flush counts of AML adults along 100 m × 1 m transects (*n* = 10) in a wheat field on the farm Kromvlei, Bothaville district, between June 1984 and December 1986. Months with zero locust counts are shown as dots. Months when no counts were made have no data.

## Data Availability

Literature and databases of historical locust outbreaks are available at the ARC-Plant Health and Protection, Pretoria.

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
