# Peer review of "Invasions and Local Outbreaks of Four Species of Plague Locusts in South Africa: A Historical Review of Outbreak Dynamics and Patterns"

_insects, 2023, doi:10.3390/insects14110846_

Round 1

Reviewer 1 Report

Comments and Suggestions for Authors

The paper is devoted to a major scientific problem of plague locust outbreaks and their control in South Africa. It perfectly suits the journal scope. As many as four major pests are covered by the paper, which makes it interesting and novel as compared to other recently published reviews. The paper is written is sound scientific language which needs only slight polishing, as listed below.

On the other hand, numerous periods of text lack respective references, including those found in Lines 91-101, 107-108,140-143, 185-188, 280-287, 298-304, 308-312, 312-325, 396-413, 442-452, 492-503, 531-548, 549-569, 586-600, 607-615, 643-652, 663-667, 687-693, 711-725, 753-757, 780-784, 785-792, 797-805, 814-825. 889-893.

Alternatively, long consequent periods of text all refer to a single source, like those in 331-352, 355-367, 371-390, citing but reference #4.

The paper pretends to belong to the review type, which makes “Materials and Methods” & “Results” sections unnecessary, further confirmed by the fact that no research approaches, other that reviewing the literature and database sources, are listed in M&M. At the same time, the paper presents numerous maps and graphs which look like original data sets (Figs. 2-9) as there are no references to the literature sources, but the methods these figures were obtained with remain totally obscure. Sometimes these methods are mentioned in the text of Results (see lines 551-569), which makes the structure of the paper even more complicated for comprehension.

Further, certain phrases directly repeat text already published by the very author (https://doi.org/10.3390/agronomy11112212), including those found in line 59-61, 64-66, 73-75, 83-84.

The manuscript should be carefully revised, taking into consideration these flaws.

Small corrections are as follows.

L37: into prevent = to prevent

L48: incomplete parentheses

L56: other grasshoppers = other species

L58: spp. and = spp., and (remove italics here as well)

L92: as aqueous = as an aqueous

L93: but then later = and later

L96 vs L468: either “motorised” or “motorized” (and take care of other verb cases)

L111: Africa, was = Africa was

L158 and further: the egg populations = the egg masses

L179: favouourable = favorable

L251 and further: “eight (8) year” – why using this double numbering system?

L260: years within this sequenced – the syntaxis is not clear

L321: generation of swarms tend = generation of swarms tends

Comments on the Quality of English Language

Small typos and impurities are listed above, slight polishing is needed

Author Response

Author response to comments and inputs made by Reviewer 1 are below in red font. The author wishes to thank the Reviewer 1 for making sound comments that certainly improve the manuscript.

1. The paper is devoted to a major scientific problem of plague locust outbreaks and their control in South Africa. It perfectly suits the journal scope. As many as four major pests are covered by the paper, which makes it interesting and novel as compared to other recently published reviews. The paper is written is sound scientific language which needs only slight polishing, as listed below.

2. On the other hand, numerous periods of text lack respective references, including those found in Lines 91-101, 107-108,140-143, 185-188, 280-287, 298-304, 308-312, 312-325, 396-413, 442-452, 492-503, 531-548, 549-569, 586-600, 607-615, 643-652, 663-667, 687-693, 711-725, 753-757, 780-784, 785-792, 797-805, 814-825. 889-893. References have now been included to address the concerns on most of the above lines.

3. Alternatively, long consequent periods of text all refer to a single source, like those in 331-352, 355-367, 371-390, citing but reference #4. Additional references and sources have now been included.

4. The paper pretends to belong to the review type, which makes “Materials and Methods” & “Results” sections unnecessary, further confirmed by the fact that no research approaches, other that reviewing the literature and database sources, are listed in M&M. At the same time, the paper presents numerous maps and graphs which look like original data sets (Figs. 2-9) as there are no references to the literature sources, but the methods these figures were obtained with remain totally obscure. Sometimes these methods are mentioned in the text of Results (see lines 551-569), which makes the structure of the paper even more complicated for comprehension.

The short Materials and Methods section was included to briefly describe where the historical locust outbreak information was sourced. It was also important to describe the changed names of the Provinces and Districts to give context. 

The manuscript is mainly a review, but does include original data sets to illustrate and describe the outbreak patterns. Some of these data should have been published years ago, but the current manuscript now offers an opportunity to make this unique and interesting historical information available.

5. Further, certain phrases directly repeat text already published by the very author (https://doi.org/10.3390/agronomy11112212), including those found in line 59-61, 64-66, 73-75, 83-84. These sentences have been changed and shortened to avoid too much repetition. 

6. The manuscript should be carefully revised, taking into consideration these flaws. Done.

7. Small corrections are as follows. Corrections have been attended to.

Reviewer 2 Report

Comments and Suggestions for Authors

This is a detailed paper providing a historical review of outbreaks of 4 species of locusts in South Africa, with a basic description of each species' biology and a detailed summary of history of outbreaks and control efforts. It concludes with perspectives on future threats.

However, I feel the text is too descriptive and does not provide much analysis of the processes involved in outbreaks or synthesis of the extensive historical data outlined that could lead to new insights.  For instance, extensive historical data on locust upsurges and migrations during past outbreaks is presented: would it be possible to do a quantitative analysis to gain a better understanding of ecological processes underlying the observed patterns?  The paper gives an excellent gives historical overview of locust control methods, but doesn't give much understanding of ecological mechanisms underlying locust outbreaks that could help plan sustainable management. 

It would help the reader to end introduction with a goal or research question: what is the research gap addressed?Please clarify the link between describing historical outbreaks and understanding current threats.

The figures are not very informative and appear to have been copied from historical documents. It would be helpful to show more detailed maps outlining the regions described in the text, superposing the information on locust upsurges and movements on  a background of ecozones and natural vs crop areas. Much descriptive geographical information in the text would be easier to understand from a map. 

Specific points:

l.185-187: mechanisms unclear - is it good rains that cause local population upsurges in diverse areas?

l. 191 What characterizes 'hotspot' farms?

l.197-199: Idea of incipient 'quality' whether genetic, or maternal effect, or sublethal pathogen infection is an interesting potential avenue for research that could be discussed further.

l.227-230: please clarify - does this mean that the cycle is shorter since control began? or outbreaks are more intense?  This could be important in understanding ecosystem impacts of control. If control is limiting potential of natural enemies to decrease populations, this is of strategic importance. 

l.244-245: this is an important point. It suggests that the control strategy needs to be rethought as it is costly and potentially of limited effectiveness. 

l.252-263: could this be shown in a figure rather than text? It's unwieldy.   

l.262-263: what are climate disruptions predictions for the region? How are they likely to impact outbreaks?

l.262-263 and l.266-267 appear contradictory: are droughts followed by outbreaks or periods of low activity?

Is data quality good enough (e.g. spatio-temporal resolution of weather data and locust counts) for a more quantitative analysis? the review mentions several relevant hypotheses but it would be nice to test them. 

Are Fig 2-5 reprinted from 1980s publication? if so, please give reference. Quality is poor, resolution low and difficult to interpret.  A single synthesis figure bringing together the main points would be more helpful.

l.402-403: does the system simply need to be modernized or rethought?  is control effective? Are side-effects on other biodiversity important? How will climate change affect rainfall dynamics in the region and hence locust dynamics? Conclusions seem superficial. Long-term sustainability cannot only be considered in financial terms, but also in environmental ones. 

l.443: 'disapiated' - a typo?

l.436-452: reads like a rather unwieldy list, this information would be better presented on a map, and the text focused on analysis and interpretation.

l.507-508: important insight, please develop.

l.547: this is important: is there potential for biocontrol in winter? use of trap crops?

l. 553: 'were'

l.762: 'warms' - typo?

l. 808, 878: 'hoper' - typo?

Author Response

Author responses to Reviewer 2 are made in red font for clarity. The author wishes to thank Reviewer 2 for important comments that improve the manuscript.

This is a detailed paper providing a historical review of outbreaks of 4 species of locusts in South Africa, with a basic description of each species' biology and a detailed summary of history of outbreaks and control efforts. It concludes with perspectives on future threats.

1. However, I feel the text is too descriptive and does not provide much analysis of the processes involved in outbreaks or synthesis of the extensive historical data outlined that could lead to new insights. 

The author has not worked on locust for over 20 years and the current manuscript was an attempt to review and publish the unique historical information before the author retires.

2. For instance, extensive historical data on locust upsurges and migrations during past outbreaks is presented: would it be possible to do a quantitative analysis to gain a better understanding of ecological processes underlying the observed patterns?  The paper gives an excellent gives historical overview of locust control methods, but doesn't give much understanding of ecological mechanisms underlying locust outbreaks that could help plan sustainable management.

The comments by the Reviewer are very true. In light of the draft manuscript being presented to the South African Department of Agriculture, it became evident that they now consider the wealth of historical information on the locust outbreaks, and especially for the brown locust, to be a vital asset for inclusion in a new expertise center being set up for ensuring the future bio security of South Africa. In this regard they have very recently provided project funding for the digitization of the historical brown locust outbreak information into a modern database and GIS format. A student will now be employed on the project to assist with the GIS mapping and the correlation of the digitized locust outbreak location data with weather patterns, especially the El Nino and La Nina oscillations, land use maps, vegetation type distribution maps, soil types, etc. At long last we can look forward to the detailed analysis of this invaluable historical data set. I certainly hope that the analysis will help us better understand the ecological mechanisms underlying the locust outbreaks.

3. It would help the reader to end introduction with a goal or research question: what is the research gap addressed? Please clarify the link between describing historical outbreaks and understanding current threats.

Valuable comment. A paragraph has now been included to describe the link between the historical information and how this could benefit the prediction and understanding of new locust threats.

4. The figures are not very informative and appear to have been copied from historical documents. It would be helpful to show more detailed maps outlining the regions described in the text, superposing the information on locust upsurges and movements on  a background of Eco zones and natural vs crop areas. Much descriptive geographical information in the text would be easier to understand from a map. The issue of these historical maps is discussed below.

Specific points:

l.185-187: mechanisms unclear - is it good rains that cause local population upsurges in diverse areas? Addressed in the text.

  1. 191 What characterizes 'hotspot' farms? Addressed in the text.

l.197-199: Idea of incipient 'quality' whether genetic, or maternal effect, or sublethal pathogen infection is an interesting potential avenue for research that could be discussed further. This interesting theory certainly needs to be investigated further. Statement made in the text that this issue needs to be investigated.

l.227-230: please clarify - does this mean that the cycle is shorter since control began? or outbreaks are more intense?  This could be important in understanding ecosystem impacts of control. If control is limiting potential of natural enemies to decrease populations, this is of strategic importance. Addressed in text.

l.244-245: this is an important point. It suggests that the control strategy needs to be rethought as it is costly and potentially of limited effectiveness. Addressed in section about the threat of the brown locust to agriculture.

l.252-263: could this be shown in a figure rather than text? It's unwieldy.   Unfortunately, there is no time to develop a figure to depict these outbreaks as the figure should also possibly show corresponding El Nino and sunspot activity.

l.262-263: what are climate disruptions predictions for the region? How are they likely to impact outbreaks? Addressed in the section about the threat of the brown locust to agriculture.

l.262-263 and l.266-267 appear contradictory: are droughts followed by outbreaks or periods of low activity? Addressed in text.

Is data quality good enough (e.g. spatio-temporal resolution of weather data and locust counts) for a more quantitative analysis? the review mentions several relevant hypotheses but it would be nice to test them. Agreed. However, this will have to be tested in a future research project once the historical data can be digitized.

Are Fig 2-5 reprinted from 1980s publication? if so, please give reference. Quality is poor, resolution low and difficult to interpret.  A single synthesis figure bringing together the main points would be more helpful. These historical maps have not been previously published in the literature. They were specifically drawn as requested by the Department of Agriculture in order to depict the course of the seasonal outbreaks. The annual maps and outbreak databases from 1980 to 2005 then became too cumbersome to be published in a Journal. The current images are high definition copies of the original hand drawn maps. 

l.402-403: does the system simply need to be modernized or rethought?  is control effective? Are side-effects on other biodiversity important? How will climate change affect rainfall dynamics in the region and hence locust dynamics? Conclusions seem superficial. Long-term sustainability cannot only be considered in financial terms, but also in environmental ones. Addressed in the section on threat of the brown locust to agriculture.  

l.443: 'disapiated' - a typo? Addressed

l.436-452: reads like a rather unwieldy list, this information would be better presented on a map, and the text focused on analysis and interpretation. Not addressed at the moment. 

l.507-508: important insight, please develop. Addressed.

l.547: this is important: is there potential for biocontrol in winter? use of trap crops? The grass headlands could be possibly be used as a ‘trap crop’ if sprayed with a stomach acting insecticide. This would need to be coordinated with the farmers and the locust officers when locust numbers were high enough to warrant the expense.

  1. 553: 'were' corrected

l.762: 'warms' - typo? corrected

  1. 808, 878: 'hoper' - typo? corrected

Round 2

Reviewer 1 Report

Comments and Suggestions for Authors

The paper is slightly improved and can now be published

Reviewer 2 Report

Comments and Suggestions for Authors

The edits have improved the paper.  My remaining concern is that source data for the figures be clearly given: on some figures (e.g. fig 2) there appears to be a legend in the bottom left cover, but it is unreadable in my copy of the text. Some later figures show no indication of the source of the information.